# Decoupling kinematics and mechanics reveals coding properties of trigeminal ganglion neurons in the rat vibrissal system

Nicholas E Bush[1†], Christopher L Schroeder[2†], Jennifer A Hobbs[3], Anne ET Yang[4], Lucie A Huet[4], Sara A Solla[3,5], Mitra JZ Hartmann[2,4*]

[1]Interdepartmental Neuroscience Program, Northwestern University, Evanston, United States; [2]Department of Biomedical Engineering, Northwestern University, Evanston, United States; [3]Department of Physics and Astronomy, Northwestern University, Evanston, United States; [4]Department of Mechanical Engineering, Northwestern University, Evanston, United States; [5]Department of Physiology, Northwestern University, Chicago, United States

**\*For correspondence:** hartmann@northwestern.edu

[†]These authors contributed equally to this work

**Competing interests:** The authors declare that no competing interests exist.

**Abstract** Tactile information available to the rat vibrissal system begins as external forces that cause whisker deformations, which in turn excite mechanoreceptors in the follicle. Despite the fundamental mechanical origin of tactile information, primary sensory neurons in the trigeminal ganglion (Vg) have often been described as encoding the kinematics (geometry) of object contact. Here we aimed to determine the extent to which Vg neurons encode the kinematics vs. mechanics of contact. We used models of whisker bending to quantify mechanical signals (forces and moments) at the whisker base while simultaneously monitoring whisker kinematics and recording single Vg units in both anesthetized rats and awake, body restrained rats. We employed a novel manual stimulation technique to deflect whiskers in a way that decouples kinematics from mechanics, and used Generalized Linear Models (GLMs) to show that Vg neurons more directly encode mechanical signals when the whisker is deflected in this decoupled stimulus space.

## Introduction

Rats, like many rodents, rely heavily on tactile information from their vibrissae (whiskers) to explore their world. Tactile signals are generated both during active whisker movement – when the rat brushes and taps its whiskers against objects – and during passive contact. Deformations of the vibrissae are transduced by mechanoreceptors in the follicle (*Ebara et al., 2002*), and the resulting electrical signals are integrated by primary sensory neurons in the trigeminal ganglion (Vg). From the Vg, signals are relayed to the brainstem trigeminal nuclei, thalamus, and primary somatosensory cortex. Neurons in the Vg are thus the 'gatekeepers' of tactile information for the vibrissal trigeminal system (*Jones et al., 2004a*; *Leiser and Moxon, 2006*, *2007*).

Several studies have demonstrated that rodents can use their vibrissae to localize objects with high precision (*Kleinfeld and Deschênes, 2011*; *Knutsen and Ahissar, 2009*; *Knutsen et al., 2006*; *Krupa et al., 2001*; *Mehta et al., 2007*; *O'Connor et al., 2010*; *Pammer et al., 2013*). Accordingly, previous work has focused on quantifying the response of Vg neurons in terms of kinematic (geometric) variables of contact, including radial distance to an object, angular position, and angular velocity (*Gibson and Welker, 1983a*, *1983b*; *Jones et al., 2004a*, *2004b*; *Leiser and Moxon, 2007*;

**eLife digest** Animals must gather sensory information from the world around them and act on that information. Specialized sensory cells convert physical information from the environment into electrical signals that the brain can interpret. In the case of hearing, this physical information consists of changes in air pressure, and for vision, it is patterns of light bouncing off of objects.

Rodents rely heavily on touch information from their whiskers to explore their world. When a whisker touches an object, it deforms and bends. The first neurons to respond to whisker touch – so called primary sensory neurons – represent contact between the whisker and the object in the form of electrical signals, but exactly how they do this is unclear.

One possibility is that primary sensory neurons encode the movement of the whisker itself. Whenever a whisker touches an object, the whisker is deflected in a particular direction by a particular amount and at a particular speed. These movement-related features are referred to as the "kinematic" properties of whisker-object contact. Alternatively, these whisker sensory neurons might be more concerned with the forces at the base of the whisker caused by object contact. These forces are the "mechanical" properties of whisker-object contact.

Bush, Schroeder et al. set out to determine whether the electrical response of these whisker sensory neurons mainly encode kinematic or mechanical information. However, these two types of information are often closely related to each other: put simply, small whisker movements tend to accompany small forces and vice versa. Bush, Schroeder et al. therefore devised a method to deliver touch stimuli to the whiskers in a way that separates kinematic from mechanical information. Mathematical models were then developed to compare how well the neurons represent each type of information. The models showed that whisker sensory neurons generally encode mechanical signals more directly than kinematic ones.

This information adds to our understanding of how animals learn about the world through their senses. However, the analysis of Bush, Schroeder et al. relies on the long-standing simplification that whisker motion is two-dimensional, whereas in reality whiskers move in three dimensions. Therefore, a future challenge is to examine how sensory neurons represent information about touch, such as the location or shape of an object, during three-dimensional whisker-object contact.

*Lichtenstein et al., 1990*; *Lottem and Azouz, 2009*, *2011*; *Lottem et al., 2015*; *Shoykhet et al., 2000*, *2003*; *Szwed et al., 2003*, *2006*).

An alternative possibility is that Vg neurons relay a high fidelity encoding of whisker mechanics – forces and moments at the base of the whisker – to be processed at later stages of the trigeminal pathway. If Vg neurons were to encode kinematic variables, a transformation from mechanical variables at the base of the whisker into kinematic variables would have to occur within the follicle (*Whiteley et al., 2015*) and/or through the primary afferent integration of mechanoreceptor responses.

Here, we directly address the question of whether Vg neurons represent mechanical or kinematic variables. It is challenging to disentangle these alternatives because the kinematics and mechanics of contact are tightly coupled under most standard experimental protocols; this coupling is especially strong during small angle deflections and when deflections occur near the whisker base. To date, this intrinsic coupling and the absence of mechanical modeling have prevented a quantitative evaluation of the extent to which Vg neurons respond to kinematic vs. mechanical inputs.

In the present study, we developed a novel manual stimulation technique that allowed us to impose large angle deflections far from the whisker base, and thereby to systematically explore large regions of the tactile input space in which mechanics and kinematics decouple. We recorded from single Vg neurons in both anesthetized and awake animals, extracted the kinematics of contact from high-speed video, and computed the mechanics of contact using a quasi-static model of whisker bending. We then used Generalized Linear Models (GLMs) to quantify Vg responses in terms of both sets of variables and investigate which description more accurately predicts Vg firing rate. We found that only when the input space is large and kinematics are decoupled from mechanics does mechanical information better predict firing activity for a majority of Vg neurons.

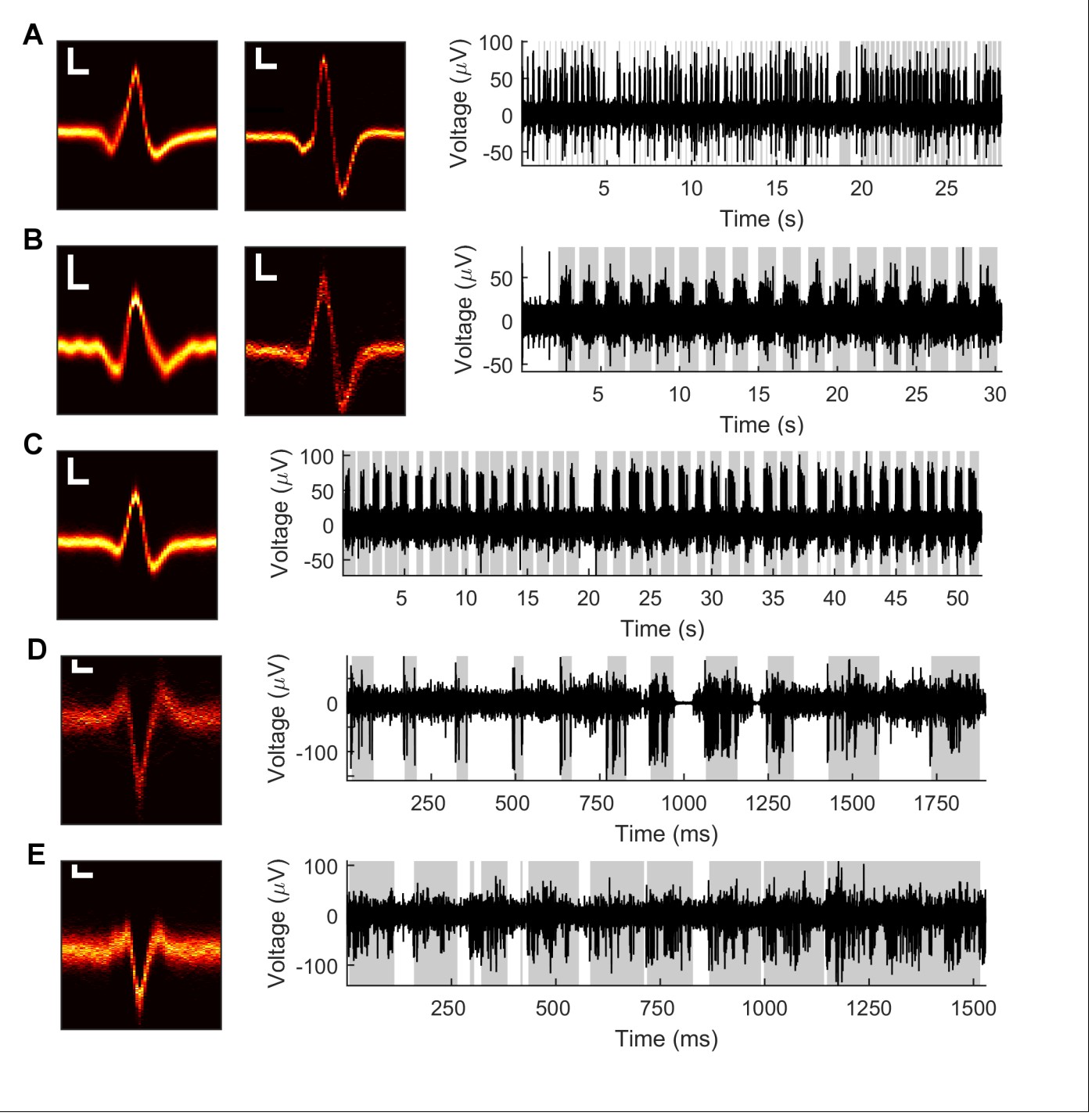

**Figure 1.** Example Vg recordings from both anesthetized and awake rats. Data from five neurons in the anesthetized animal (**A–C**) and two neurons in the awake animal (**D–E**). Left: Heatmaps of isolated spike waveforms over all recordings of each neuron. Two waveforms in **A** and **B** indicate simultaneously recorded neurons. Scale bars are 20 µV, 200 µs; width of waveforms is 1.5 ms. Right: Segments of bandpass filtered (300–6,000 Hz) raw neural traces during periods of passive deflection in the anesthetized animal (**A–C**) or active contact in the awake animal (**D–E**). Gray shading indicates periods of contact.

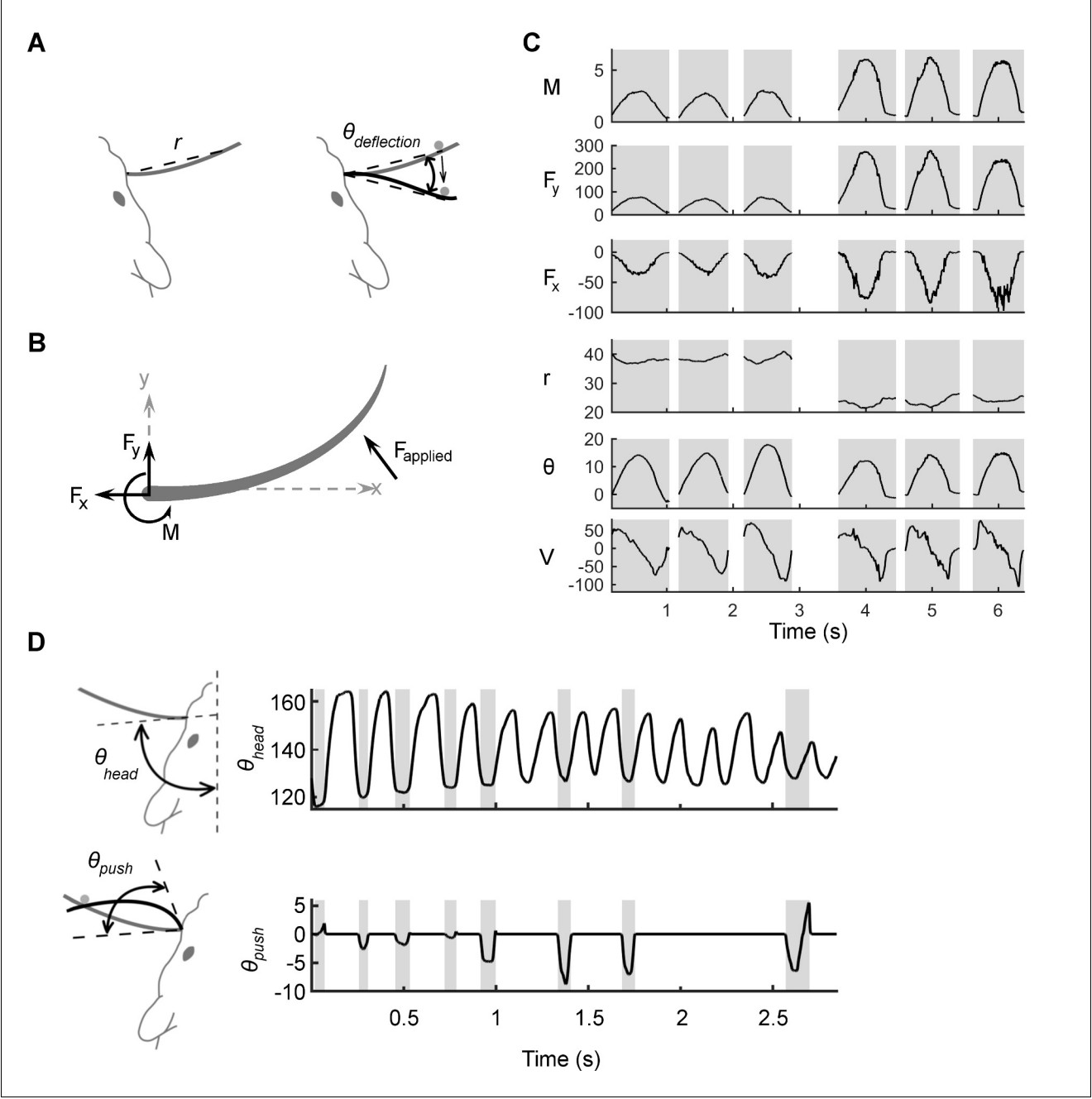

**Figure 2.** Mechanical and kinematic variables associated with contact. (**A**) Schematic of the kinematic variables of contact. The shape and position of the whisker when at rest is in gray. The variable $r$ indicates the straight-line distance from the basepoint to the contact point. During passive deflections, the relevant angle is $\theta_{\text{deflection}}$, the angle between the line segment that connects the basepoint to the current point of contact and the line segment that connects the basepoint to the initial contact point. The velocity $V$, not shown, is the temporal derivative of $\theta_{\text{deflection}}$. (**B**) Schematic of the mechanical variables of contact: bending moment ($M$), and the transverse ($F_y$) and axial ($F_x$) components of the applied force ($F_{\text{applied}}$). All variables are computed at the whisker base. (**C**) Examples of mechanical and kinematic variables during six manually delivered passive deflections in the anesthetized rat. Shading denotes contact episodes. The stimulations are similar but not identical to each other; this imparts a naturalistic variability to the tactile inputs. Units for $F_x$ and $F_y$ are µN; $M$ is in µN-m; $r$ is in mm; $\theta$  is in degrees; and $V$ is in degrees/s. (**D**) In the awake rat, $\theta_{\text{deflection}}$ is no longer well defined, and the relevant angle is $\theta_{\text{push}}$, the angle swept out by the tangent to the whisker at its base as the whisker deflects against an object. The velocity $V$ is the temporal derivative of $\theta_{\text{push}}$. The figure illustrates that $\theta_{\text{head}}$, the angle between the tangent to the whisker at its base and the midsagittal plane, is not a valid kinematic variable to explain neural responses because it varies independently of contact.

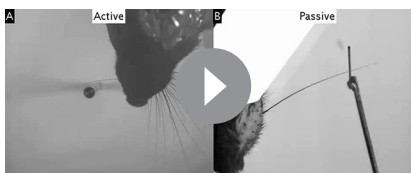

**Video 1.** Comparison of active whisking with passive, manual deflection. Two seconds of high speed video (**A**) as an awake, body restrained rat whisks against a peg, and (**B**) as the whisker is passively deflected using manual stimulation in the anesthetized animal. Videos are slowed by factors of ~16 and ~15, respectively.

# Results

## Quantifying the kinematic and mechanical variables of contact

We recorded high-speed video (300 fps) during manual deflection of 18 single whiskers in anesthetized rats while simultaneously recording neural responses from 22 Vg neurons. Example neural data are shown in *Figure 1A–C*. Whiskers were deflected with a hand-held graphite probe in two directions (rostro-caudal and caudo-rostral), with amplitudes up to several mm. Stimulation was delivered at variable radial distances that ranged up to ~90% of the whisker length, and at two speeds: 'fast' and 'slow.' Note that manual stimulation caused radial distance of contact, velocity, and deflection amplitude to vary across deflections. The two dimensional (2D) whisker shape was tracked in each video frame to quantify the kinematic and mechanical variables of contact.

Kinematic variables are illustrated in *Figure 2A* and consist of the radial distance of contact ($r$), the angular displacement ($\theta_{\text{deflection}}$), and the velocity of deflection ($V$, the temporal derivative of $\theta_{\text{deflection}}$, not shown). Kinematic variables were extracted directly from the shape of the whisker, as detailed in *Materials and methods*. During non-contact times, all kinematic variables are undefined.

The mechanical variables of contact were computed numerically based on the full tracked whisker shape using a quasi-static, frictionless model of elastic beam bending (see *Materials and methods*; [*Birdwell et al., 2007*; *Quist and Hartmann, 2012*; *Solomon and Hartmann, 2008*, *2010*]). As illustrated in *Figure 2B*, in 2D the three mechanical signals at the base of the whisker are bending moment ($M$), transverse force ($F_{\text{y}}$), and axial force ($F_{\text{x}}$). Because the mechanical model is quasi-static, all mechanical signals are exactly zero during periods of non-contact.

Examples of both mechanical and kinematic variables are shown in *Figure 2C*, which shows the signals evoked during six passive deflections of the whisker at two different radial distances. Shaded regions indicate contact episodes. Notice that each deflection varies slightly from every other deflection, reflecting the naturalistic variability of manual stimulation.

In a separate group of animals we recorded high-speed video (1000 fps) while rats explored a vertical pole (seven whiskers, nine neurons). Examples of neural data recorded in the awake animal are shown in *Figure 1D–E*. Whisker shape was tracked, and the kinematic and mechanical variables of contact were calculated. *Video 1* compares examples of manually delivered deflections and active whisking behavior.

The variables that describe active whisking are the same as those for passive contact, except that the calculation of the angular position of contact must change. In the awake animal, the contact point does not move with respect to the whisker basepoint, so $\theta_{\text{deflection}}$ is not well defined. Instead the relevant angle is $\theta_{\text{push}}$ (*Figure 2D*, bottom left), the angle swept out by the tangent of the whisker base from the time of contact onset to the current time (*Bagdasarian et al., 2013*; *Kaneko et al., 1998*; *Mehta et al., 2007*; *Quist and Hartmann, 2012*; *Solomon and Hartmann, 2006*, *2011*).

Given that the present work aims to compare the relative ability of mechanical and kinematic variables to describe Vg responses, which are strongly affected by contact, it is not appropriate to use the angle of the whisker with respect to the midsagittal plane ($\theta_{\text{head}}$) as a kinematic variable. The angle $\theta_{\text{head}}$ contains no information about contact; note in *Figure 2D* that $\theta_{\text{head}}$ varies significantly throughout the trial, while $\theta_{\text{push}}$ varies only during contact. If the variable $\theta_{\text{head}}$ were used as an input, it would unfairly favor a mechanical explanation for Vg firing because it would add a variable with no contact information to the kinematic hypothesis.

We have not included whisking phase (i.e. the relative value of $\theta_{\text{head}}$ within each whisking cycle) as a potential explanatory variable for the response of Vg neurons. Although this variable is represented in Vg responses during non-contact whisking (*Wallach et al., 2016*) and is of clear importance in central trigeminal structures (*Curtis and Kleinfeld, 2009*; *Fee et al., 1997*), the present

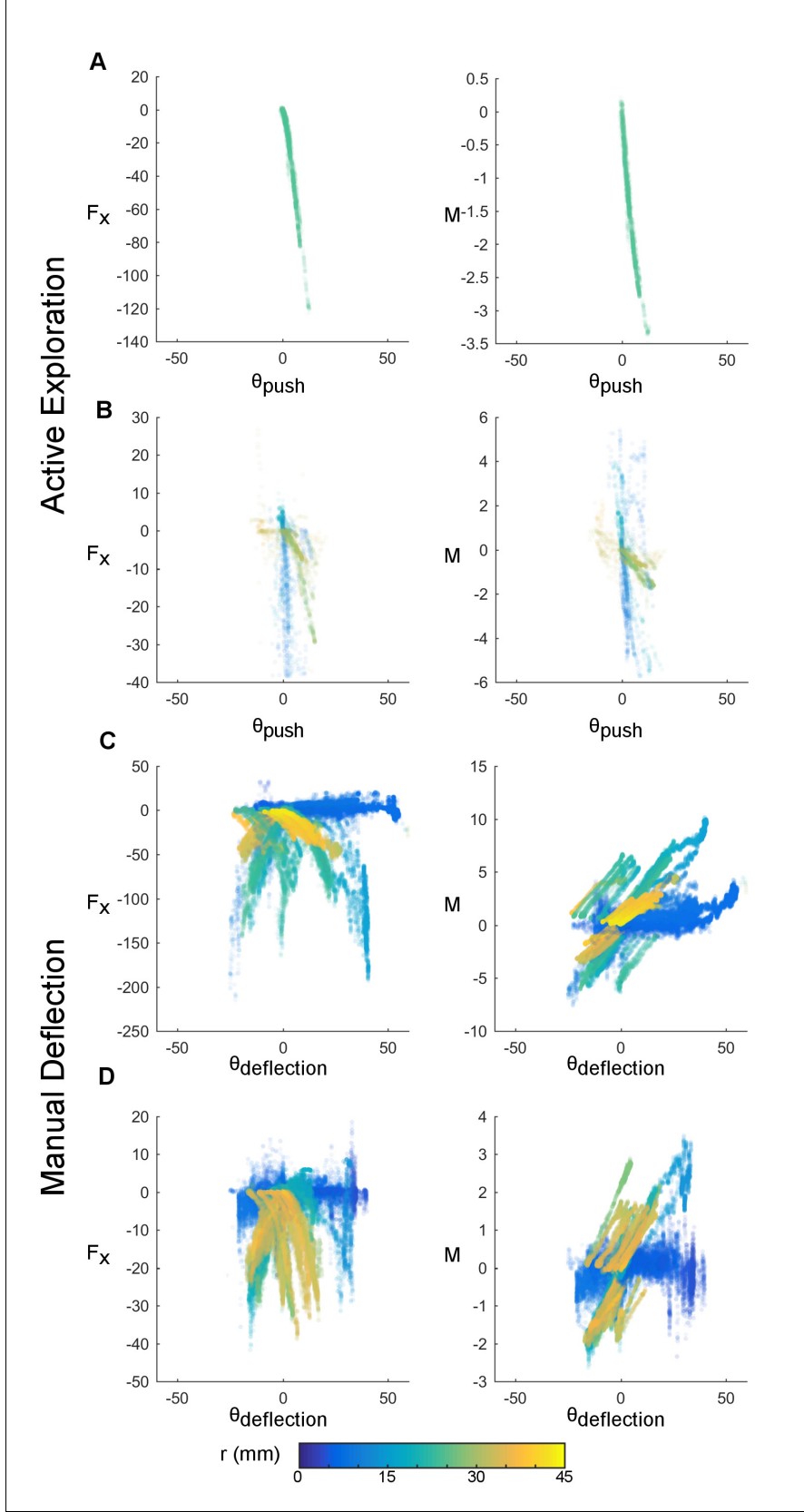

**Figure 3.** Manual stimulation reliably decouples mechanical and kinematic variables. Mechanical and kinematic variables of contact are shown across trials of active whisking (rows A and B, whiskers C1 and Gamma respectively)
*Figure 3 continued on next page*

*Figure 3 continued*

and passive manual stimulation (rows C and D, whiskers B1 and D1 respectively). Awake trials were 3.02 s (**A**) and 12.9 s (**B**) in duration; passive trials were 64.67 s (**C**) and 114.53 s (**D**) in duration. Each point represents the observed mechanical and kinematic inputs for a 1 ms time bin. The x-axis depicts the angular coordinate of contact in degrees, the y-axis either the axial force ($F_x$, units of µN) or moment ($M$, units of µN-m). Color represents the radial distance of contact in mm. During manual deflection, a larger input space is sampled. The actual range spanned by the mechanical variables depends on whisker identity.

study is limited to an analysis of contact whisking, during which kinematic and mechanical coding can be directly compared.

## Manual stimulation decouples kinematic and mechanical variables of contact

To determine the extent to which Vg neurons encode the mechanics or kinematics of contact, it is essential to observe contact conditions under which these two sets of input variables are decoupled. *Figure 3* compares kinematic and mechanical variables computed for two whiskers during active exploration (*Figure 3A–B*) to those observed during passive, manual deflection (*Figure 3C–D*).

Mechanical and kinematic variables are often tightly coupled during awake behavior (*Figure 3A*). Although some degree of decoupling is possible in the actively whisking animal (*Figure 3B*), the explored regions in input space depend on the animal's behavior. It is challenging to reliably sample a large, decoupled input space with the awake animal.

In contrast, manual stimulation offers a simple and reliable method to explore a large, decoupled region of the input space (*Figure 3C–D*). Manual stimulation can involve large angle deflections (up to 60°) at large radial distances (up to 45 mm) more consistently than in the actively behaving animal. Exploring these large regions decouples the kinematic and mechanical inputs, allowing us to address the question of whether Vg neurons encode mechanics or kinematics.

*Videos 2–5* show rotating views of three dimensional versions of the plots in *Figure 3*, now including the radial distance of contact $r$ as a third axis.

## Follicle state in the awake and anesthetized animal

It is possible that the rigidity with which the whisker base is held during contact differs between the awake and anesthetized animal. In the awake animal, capillaries at the level of the cavernous sinus could increase hydrostatic pressure and thereby the rigidity of the whisker-follicle junction (*Rice, 1993*). In addition, the activation of muscles surrounding the whiskers could increase the rigidity of the follicle with respect to the mystacial pad. Either or both of these changes near the whisker base could alter the whisker's deformation in response to an applied force. Given that the follicle-whisker junction has been shown to be rigid in the anesthetized animal (*Bagdasarian et al., 2013*), blood-based hydrostatic changes are unlikely to be responsible for differences in rigidity between awake and anesthetized states. Changes in muscle activation, however, are a potentially significant effect that remains to be fully investigated.

In the anesthetized animal, we observed large translations and rotations of the follicle in the skin when a force is applied to the stiff, proximal portion of the whisker (*Video 6*). Translations and rotations were not observed during contacts at the more flexible, distal portion of the whisker; this rigidity is similarly observed in the awake animal, where mystacial muscles prevent movement of the follicle during contact.

We therefore restricted our analyses in the anesthetized animal to distal contacts ( ≥40% of the whisker length), where the apparent rigidity of the whisker-follicle-skin interface is significantly greater than the rigidity of the whisker at contact and the follicle does not move appreciably during contact.

## Generalized linear models

We employed generalized linear models (GLMs) to determine the relative importance of kinematic and mechanical variables in predicting neural firing. GLMs include linear combinations of the history of various input variables, as well as the non-linear characteristic of biological neurons, to predict the

**Video 2.** 3D visualization of mechanical and kinematic relationships for one neuron recorded in the awake animal. Rotating view of inputs to the neuron shown in *Figure 3A*, active exploration. Radial distance is represented along the third axis.

**Video 3.** 3D visualization of mechanical and kinematic relationships for a second neuron recorded in the awake animal. Rotating view of inputs to the neuron shown in *Figure 3B*, active exploration. Radial distance is represented along the third axis.

firing rate of a neuron given previously observed stimulus inputs and the resultant spiking patterns (*Pillow et al., 2008*). The GLM approach lends itself to the analysis of both active and passive deflections. 'Full model' GLMs were constructed using the three mechanical and the three kinematic variables ($F_y$, $F_x$, $M$, $r$, $\theta$, $V$) as input variables (predictors) for the observed spike train at 1 ms resolution.

We invoke a formulation of the GLM in which the predictors are convolved with a set of nonlinear basis functions ('raised cosine bumps') that cover a desired temporal window into the past over which to consider the stimulus history (*Pillow et al., 2008*). Here, we choose the five dimensional basis shown in *Figure 4A*. Each predictor thus gives rise to five 'convolved predictors', each with the temporal structure of the corresponding basis function. The basis functions extended 75 ms into the past, to match the temporal extent of the cross-correlations between the observed spikes and the various predictors while not being longer than the shortest inter-stimulus interval.

This procedure gives us a total of 30 'convolved predictors' (6 predictors * 5 basis functions) that are the inputs to the model. The GLM then fits optimal coefficients ($\beta_{lj}$, $1 \leq l \leq 5$, $1 \leq j \leq 6$) for each of the 30 convolved predictors, where $l$ is the index of the basis function and $j$ is the index of the predictor. The model includes one additional coefficient $\beta_0$ for a constant term. These 31 coefficients are used to construct a linear combination of the 30 convolved predictors; this linear combination is the argument to a sigmoidal nonlinearity that outputs the instantaneous probability of firing at every 1 ms time bin.

Before convolving with the basis set, the predictors are whitened to have zero mean and unit standard deviation. This allows us to compare $\beta$ coefficients for different predictors that would otherwise be on different scales. *Figure 4B* shows the mean absolute value of the $\beta$ coefficients across all neurons. Each set in this figure refers to a particular basis function; the coefficients labeled as $\beta_1$ actually comprise all six coefficients $\beta_{1j}$, $1 \leq j \leq 6$, where the index $j$ labels the predictors ($F_y$, $F_x$, $M$, $r$, $\theta$, $V$). The six coefficients labeled as $\beta_1$ represent the weight of the most temporally recent and precise time period as specified by the basis function $b_1$; this period covers 0 to 4 ms into the past with a peak time at 0 ms. The most recent time period is clearly the most important in predicting spikes for all six predictors. Subsequent sets of coefficients represent the importance of more distant past times, as specified by the corresponding basis functions shown in *Figure 4A*. The very small values of the coefficients $\beta_5$ associated with the basis function $b_5$ indicate that there is no

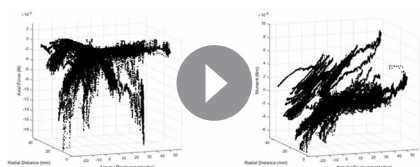

**Video 4.** 3D visualization of mechanical and kinematic relationships for one neuron recorded in the anesthetized animal. Rotating view of inputs to the neuron shown in *Figure 3C*, manual deflection. Radial distance is represented along the third axis.

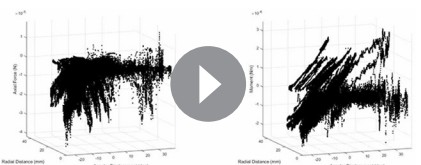

**Video 5.** 3D visualization of mechanical and kinematic relationships for a second neuron recorded in the anesthetized animal. Rotating view of inputs to the neuron shown in *Figure 3D*, manual deflection. Radial distance is represented along the third axis.

need to look much further than 25 ms into the past. Among all predictors, moment $M$ has the largest coefficient $\beta$ for the first four basis functions; this indicates that on average, moment is the most important predictor of firing activity.

As detailed in *Materials and methods*, it is useful to obtain predictor specific filters $\alpha_j$, $1 \leq j \leq 6$, as a linear combination of the basis functions $b_l$, $1 \leq l \leq 5$, with the coefficients $\beta_{lj}$, $1 \leq l \leq 5$, $1 \leq j \leq 6$ obtained from the GLM fit. These predictor-specific filters, shown in *Figure 4C* for an example neuron, illustrate the impact of each predictor on the neuron's firing. Note that the filters shown in *Figure 4C* decay to zero after about 15 ms, and that for this neuron, a change in moment from negative to positive, a negative $\theta$, and a negative $F_y$ are the inputs that drive the cell to fire. An alternative characterization of inputs relevant to Vg firing follows from calculating spike-triggered averages (STA) for each of the input variables. The STAs for the neuron depicted in *Figure 4C* are shown in *Figure 4—figure supplement 1*.

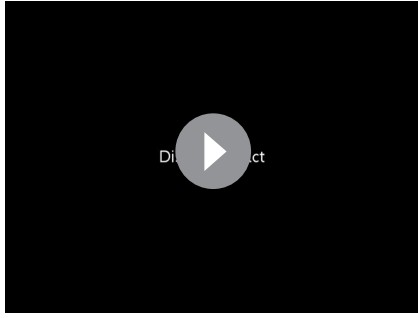

**Video 6.** Comparison of distal and proximal contact in the anesthetized rat. High speed video of distal and proximal contacts (3 s per clip, slowed by a factor of ~5) highlights the movement of the follicle relative to the skin when contact is made close to the whisker base.

## Relative importance of predictor variables

Bending moment is not only the most important input to the example neuron in *Figure 4C*, but also emerges as the most important input across all neurons in *Figure 4B*. However, all input variables contribute to the GLM fits. Different neurons might respond strongly to different combinations of input variables. To quantify whether kinematic or mechanical variables provide better predictions of firing activity, we constructed separate GLMs that had access to only the kinematic variables or only the mechanical variables. We refer to these models as 'subset models'. We calculated the coefficient of determination ($R^2$) between the predicted spiking probability given by these subset models and the predicted spiking probability of the full model. Note that this metric is not a measure of how well the models predict the neuron's firing, but rather of how much of the information captured by the full model can be accounted for by either of the two subset models.

Examples of the relationship between the subset model predictions and the full model predictions are shown in *Figure 5A*. For neuron 24, the predictions of the mechanical subset model correspond well to those of the full model ($R^2 = 0.88$), while the predictions of the kinematic subset model do not ($R^2 = 0.08$). This result indicates that the information present in the mechanical variables accounts for most of the information that the full model uses to predict spike rates. The opposite is true for neuron 8: the information present in the kinematic variables better accounts for the information that the full model uses to predict spike rates.

The quality of the subset models is quantified over all neurons in *Figure 5B*, which plots the $R^2$ values between the predictions of the mechanical subset model and those of the full model against the $R^2$ values between the predictions of the kinematic subset model and those of the full model. An inverse relationship is apparent, indicating that if the predictions of one subset model account well for the predictions of the full model, the predictions of the other subset model do not.

So far, our analysis has not addressed the quality of the full model predictions. To quantify the accuracy of the full model, we computed the Pearson Correlation Coefficient ($R$) between the GLM predicted rate and the observed spike rate, obtained by smoothing the spike train with a Gaussian kernel ($\sigma = 15$ ms; see *Materials and methods*). In *Figure 5B*, data points are shaded red if their $R$ is above the median $R$ value (0.3), and grey if their $R$ is equal to or below the median $R$ value. A majority of red markers (10/15) fall below the diagonal, suggesting that when the full model relies on the information provided by the mechanical subset of input variables, the model performs better.

We next asked how well the full model and the subset models could predict the spike rate of each neuron. The distribution of $R$ values for the full model is shown in *Figure 6A*. The median $R$ value across all neurons is 0.30. There was no significant difference between active contact and passive deflections (Wilcoxon rank-sum test p=0.18).

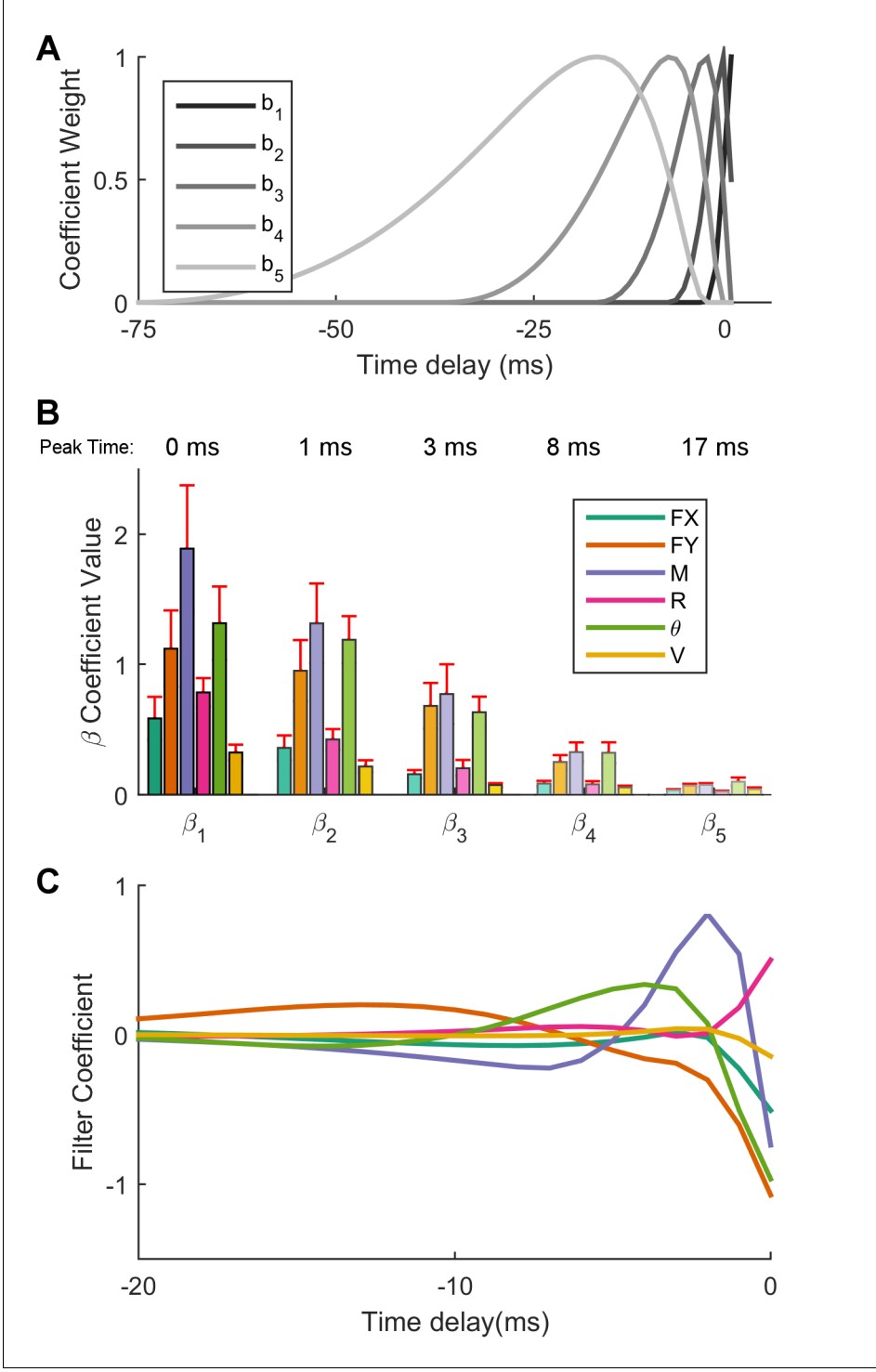

**Figure 4.** Optimal linear filters indicate that moment is the most important predictor of Vg neural firing. (**A**) The non-linear basis of 'raised cosine bumps'. (**B**) Average absolute value of the GLM fit coefficients ($\beta$) across all neurons. $\beta_l$ refers to the coefficient of $l_{th}$ cosine basis function, with $\beta_1$ being the most recent and precise, and $\beta_5$ being the most delayed and diffuse. Shading corresponds to the basis function plotted in (**A**). Two neurons have been omitted from this aggregate analysis because their outlying coefficients $\beta$ (order $10^{13}$) distorted the averages reported here. (**C**) The linear combination of the basis functions $b_l$ plotted in (**A**) with the coefficients $\beta_{lj}$ obtained from the GLM fit allows us to obtain predictor specific filters $\alpha_j$, shown here as a function of time (truncated at 20 ms for visualization) for an example neuron. These filters quickly decay to zero, indicating that the majority of the

*Figure 4 continued on next page*

*Figure 4 continued*

information important to the cell is contained in the preceding few milliseconds. For the cell shown here, moment, transverse force, and angular displacement are important input signals, with moment being the most important.

The following figure supplement is available for figure 4:

**Figure supplement 1.** Examples of spike-triggered averages of the six input variables for the cell shown in *Figure 4C*.

We then asked how the accuracy of the subset models compares to that of the full model for both active contacts and manual deflections. In *Figure 6B* we plot the distribution of the percent error between the full model and each of the subset models. Percent errors near zero indicate that the subset model performed as well as the full model; values below zero indicate that the subset model performed better than the full model. The data shown in *Figure 6B* omit two points for which the full model performs worse than both subset models. These points also exhibited the worst full model performance, with R values smaller than 0.05. All subsequent analyses omit these two points.

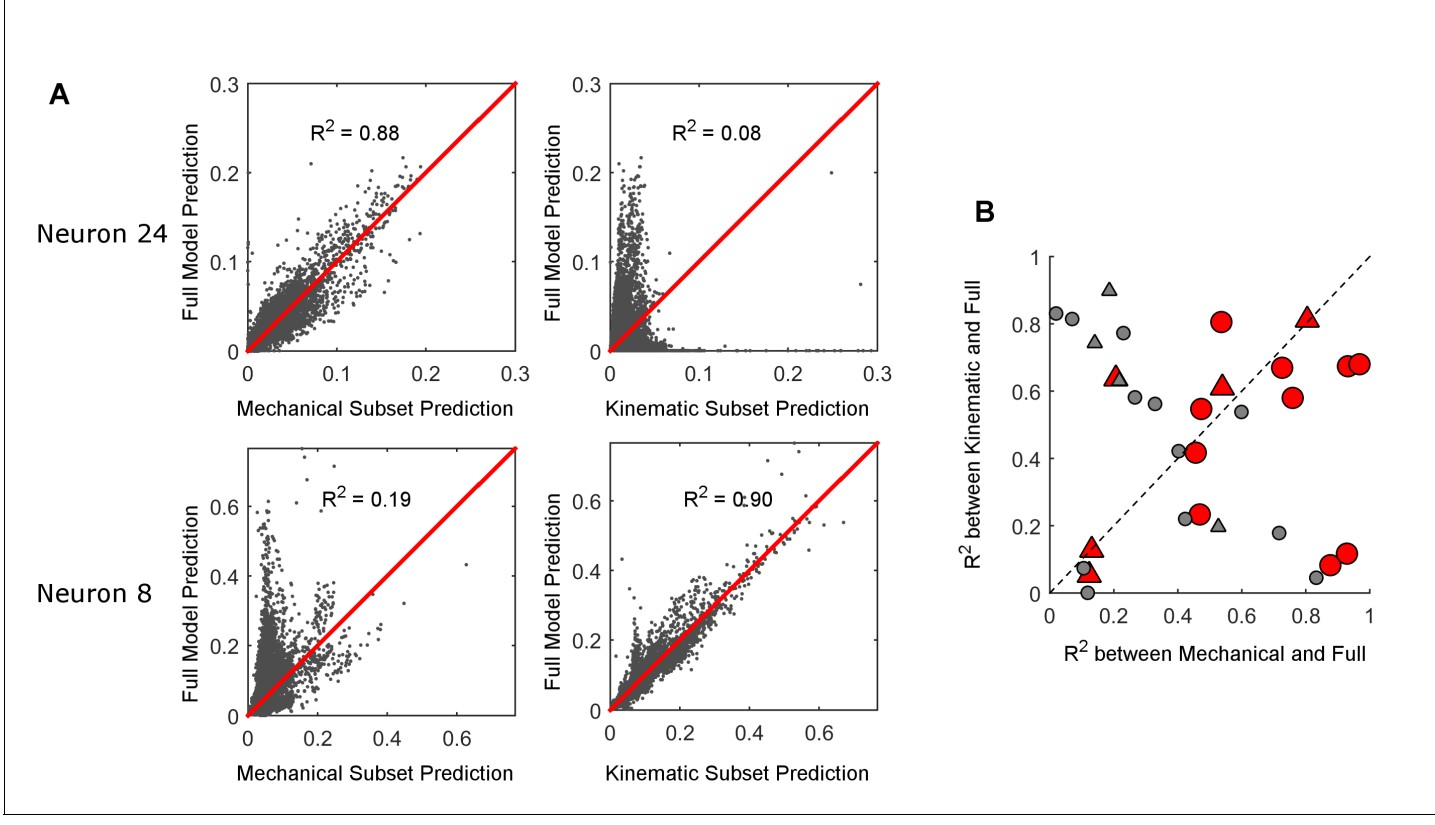

**Figure 5.** Comparison between full and subset models. (A) The firing rate prediction of each subset model is plotted against the prediction of the full model. The predictions are probability of a spike in each 1 ms time bin. For neuron 24 in the first row, the mechanical model is well correlated with the full model and the kinematic model is not; the opposite is true for neuron 8 in the second row. (B) The $R^2$ between the firing rate predicted by the full model and the firing rate predicted by each subset model (mechanical on the x-axis; kinematic on the y-axis). Each data point represents one neuron. The triangles represent neurons recorded during active contact; the circles represent neurons recorded during manual deflections. Red markers correspond to models that predict the cell's spike rate better than the median accuracy (R>0.30). Gray markers indicate poor prediction accuracy (R≤0.30).

The following source data is available for figure 5:

**Source data 1.** Summary data used to create *Figure 5B* are reported.

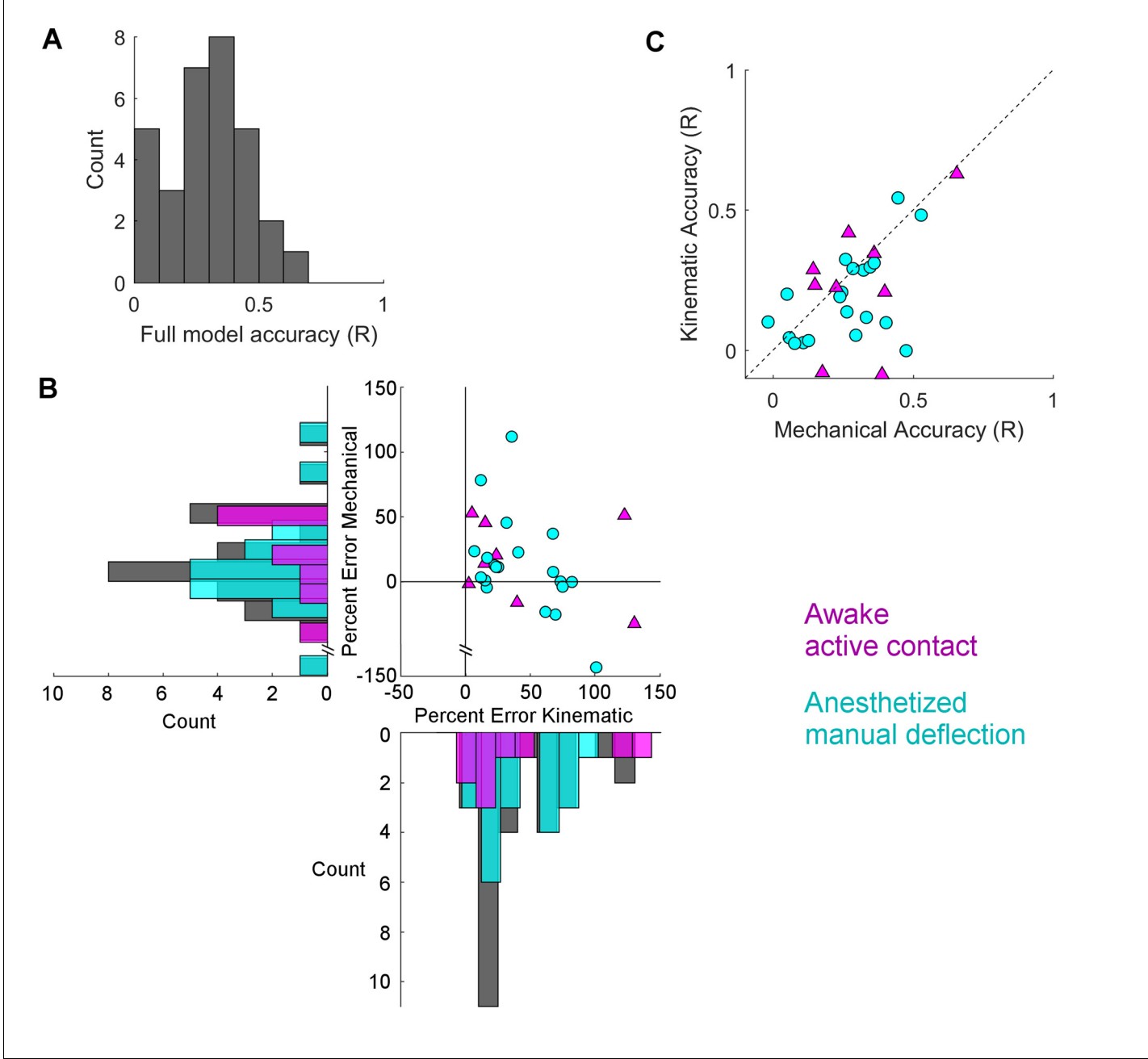

**Figure 6.** Mechanical models outperform kinematic models for manual deflections. Pearson correlation coefficients (R) between GLM predictions and observed spike rate smoothed at 15 ms are compared between the full model and the subset models. (**A**) Histogram of Pearson correlations between the spike rates predicted by the full model and the observed spike rates, for all neurons. (**B**) Percent error between the R value for the full model and for each of the subset models is plotted for each neuron. Active contact responses are plotted as magenta triangles, manual deflections as cyan circles. Values close to zero indicate that the subset model performed almost as well as the full model; values less than zero indicate that the subset model performed better than the full model. Histograms indicate the distributions of the percent differences of each subset model for active contacts (magenta), manual deflections (cyan), and the whole population (gray). For the data shown here (see text), results for the subset model trained on mechanical data are significantly closer to zero for manual deflections but not for active contacts. (**C**) The R values for the two subset models are plotted against each other. Points that lie below the diagonal indicate that the mechanical model better predicted the spike rate than the kinematic model. Color and marker scheme same as in (**B**).

The following source data is available for figure 6:

**Source data 1.** Summary data used to create *Figure 6* are reported.

For the manual stimulation data, the median percent error for the mechanical subset models tend to lie closer to zero than the median percent error for the kinematic models (Wilcoxon signed rank test p<0.05); in contrast, there is no such trend for the active contact data (Wilcoxon signed rank test p=0.43).

Finally, in *Figure 6C*, we compared the accuracy of the mechanical subset model versus that of the kinematic subset model for both active contacts and passive, manual deflections. We found that 75% (15/20) of neurons recorded with passive stimulation lie below the diagonal (linear model slope 95% CI = [0.20 0.96], paired t-test p<0.05), while those neurons recorded with active touch lie closer to the diagonal (6/9 above, 3/9 below; linear model slope 95% CI = [-0.42, 1.73], paired t-test p=0.37). These results indicate that although the mechanical model better predicts firing during manual stimulation, there is no evident preference for kinematic or mechanical models during active touch.

The input space characterization in *Figure 3* explains why it is not possible to distinguish between subset models during active contact: in this scenario, the input space is relatively small and the kinematics and mechanics tend to be more tightly coupled than under manual, passive stimulation. If the inputs to the two subset models are highly coupled – as they are in the active case – then these models receive similar input information and neither can expected to predict Vg activity better than the other.

## Discussion

Neurons of the trigeminal ganglion are the gatekeepers of all available tactile information in the rodent vibrissal system. The manner in which these neurons represent tactile information places direct constraints on the processing performed by more central trigeminal structures, including thalamus and cortex. Understanding how information is encoded and transformed in Vg neurons is thus essential to obtaining an understanding of vibrissal related responses in these central structures.

### Kinematic and mechanical signals are coupled in standard passive stimulation experiments

Historically, responses of neurons in the vibrissotrigeminal system have been described in terms of whisker kinematics (*Gibson and Welker, 1983b*; *Jones et al., 2004a*, *2004b*; *Leiser and Moxon, 2007*; *Lottem et al., 2015*; *Moore et al., 2015*; *Shoykhet et al., 2000*; *Simons, 1978*; *Szwed et al., 2003*, *2006*; *Zucker and Welker, 1969*). More recently, however, studies have suggested that mechanics offer an alternative explanation for firing properties of neurons at multiple levels of the trigeminal pathway (*Campagner et al., 2016*; *Chen et al., 2015*; *Hires et al., 2015*; *Xu et al., 2012*).

The possibility that Vg neurons encode the mechanics of touch is not inconsistent with the body of literature describing kinematic encoding, because mechanical and kinematic variables are often inherently coupled. It is common to stimulate whiskers through small angles close to the base, so that almost no bending of the whisker occurs (*Gibson and Welker, 1983b*; *Jones et al., 2004a*, *2004b*; *Lichtenstein et al., 1990*; *Zucker and Welker, 1969*). Under these stimulation conditions there is no room for mechanics and kinematics to decouple, making it impossible to distinguish between these two coding possibilities. *Campagner et al. (2016)* elegantly demonstrate this coupling during passive stimulation with a piezoelectric (piezo) bender. They show that during piezo stimulation, curvature change and angle are tightly correlated; GLMs based on either of these variables therefore produce indistinguishable predictions. They further show that in the awake animal, in contrast, curvature change and angle are decorrelated; they attribute this decorrelation to the awake condition.

The novel manual stimulation paradigm of the present work demonstrates that kinematics and mechanics are not necessarily coupled during passive stimulation, nor necessarily decoupled during active contact (*Figure 3*). Decoupling is essential to distinguish between the two possible coding schemes in the trigeminal ganglion.

## Decoupling mechanical and kinematic signals during contact reveals that Vg responses are better predicted by mechanics than kinematics

The novel manual whisker stimulation protocol employed in the present work allows us to reliably explore larger regions of input space in which the strong coupling between mechanics and kinematics breaks down (*Figure 3*). By working in this decoupled regime, the present study demonstrates that Vg neurons more closely represent mechanical rather than kinematic variables during contact. The optimal filters produced by the GLM more heavily weight the mechanics of contact; on average, bending moment is the most important predictor in models that have access to both mechanical and kinematic inputs (*Figure 4B*).

Furthermore, in cases where mechanics (rather than kinematics) account for most of the predictive ability of the full model, the full model better predicts the spiking behavior of the neuron. The predictive accuracy of models with access to only mechanical inputs is frequently as good as that of models with access to all inputs; this is less frequently the case for models with access to only kinematic inputs (*Figure 6B*). Finally, models with access to only mechanical inputs perform better than those with access to only kinematic inputs (*Figure 6C*).

Importantly, the improved predictive accuracy attributed to mechanical variables is seen only for experiments in which the kinematics and mechanics are decoupled and thus carry distinct information. In our experiments, body-restrained awake animals only infrequently exhibited the type of whisking behavior that would be required to sample a large input space and decouple kinematics and mechanics. Accordingly, models of Vg responses in the awake animal based on mechanical variables rarely outperformed those based on kinematic variables, mirroring the null result observed by *Campagner et al. (2016)* during passive stimulation when mechanical and kinematic information were coupled.

It is worth emphasizing that our conclusions, as well as those of *Campagner et al. (2016)*, regarding the comparative ability of kinematic and mechanical variables to predict the firing of Vg neurons, are based on a simple model of neural encoding: that Vg neurons respond to a linear combination of relevant features of the stimulus, followed by a global static nonlinearity that accounts for the Poisson statistics of the spike generation process. This is the conceptual framework that underlies the choice of GLM models, whose ability to predict the firing of Vg neurons in response to passive stimulation was first established by *Bale et al. (2013)*. In asking which set of variables, kinematic vs mechanical, are better predictors of Vg activity when used as inputs to a GLM model, we ask which set of variables is more informative within the hypothesis of linear-non-linear (LNL) encoding.

## Kinematic and mechanical variables as explanatory variables for Vg firing

At first glance, some results of the present work may appear to contradict those of *Campagner et al. (2016)*. Our results show that mechanical models perform better than kinematic models in anesthetized experiments but show little distinction in the awake animal. In direct contrast, *Campagner et al. (2016)* find similar performance of mechanical and kinematic models in the anesthetized animal but that mechanical models perform better than kinematic models in the awake preparation.

The fundamental reason for the apparent discrepancy is that in the awake animal *Campagner et al. (2016)* use a kinematic variable ($\theta_{\text{head}}$) that varies independently of object contact, but a mechanical variable (change in curvature) that varies only with contact. Given that the response of Vg neurons is strongly correlated with contact (*Leiser and Moxon, 2007*; *Zucker and Welker, 1969*), the mechanical variable will necessarily have a higher predictive value, especially at 100 ms time scales that match the duration of a whisk.

The reason $\theta_{\text{head}}$ is independent of contact is that this angle is measured with respect to the midline of the animal's head. In contrast, change in curvature at the base (a proxy for bending moment) is measured independently of the whisker's position relative to the head. The angle $\theta_{\text{head}}$ and curvature change will be decoupled in the awake experiments because contact with an object can occur at different positions relative to the head. For example, a whisker can exhibit very similar curvature changes regardless of whether it makes contact with a peg at $\theta_{\text{head}} = 70°$ or at $\theta_{\text{head}} = 110°$.

In *Campagner et al. (2016) Figure 4G* it is clear that if one were to account for the value of $\theta_{\text{head}}$ at the initial contact with the pole, curvature change would be strongly correlated with an angle

that would not be $\theta_{\text{head}}$ but $\theta_{\text{push}}$ – the angle used in the present work and in other studies of mechanical coding of object location (*Bagdasarian et al., 2013*; *Birdwell et al., 2007*; *Kaneko et al., 1998*; *Pammer et al., 2013*; *Solomon and Hartmann, 2011*). *Campagner et al. (2016)* briefly address this point. Their results from the awake animal show smaller differences in performance between models based on kinematic or mechanical inputs when $\theta_{\text{push}}$ is used as the kinematic variable, consistent with the present findings.

In the anesthetized experiments of *Campagner et al. (2016)*, $\theta_{\text{head}}$ and curvature are always strongly correlated because the whisker is trimmed (to 5 mm), angles of deflection are relatively small (10°), and the deflection is always applied at the same value of $\theta_{\text{head}}$. Had these experiments used large amplitude deflections and/or deflections further from the whisker base, $\theta_{\text{head}}$ would presumably have decoupled from curvature changes.

More subtly, the quantification of mechanical inputs differs between the present work and that of *Campagner et al. (2016)*. Forces and moments at the whisker base cannot be measured directly because any sensor placed at the whisker base would interfere with the whisker's mechanics. *Campagner et al., (2016)* use curvature change at the base as a proxy for bending moment, an approximation based on linear elastic beam theory (*Beer et al., 2015*). In contrast, we use a validated quasi-static model of whisker bending to compute the forces and moments at the base during contact (*Birdwell et al., 2007*; *Huet and Hartmann, 2016*; *Huet et al., 2015*; *Solomon and Hartmann, 2008*, *2010*). This model accounts for the full shape of the whisker and offers the advantage of computing the axial and transverse forces in addition to bending moment.

## A mechanical framework for interpreting primary sensory signals during both contact and non-contact whisking

Our work and that of *Campagner et al. (2016)* agree that Vg neurons encode mechanical variables more robustly than kinematic variables; we suggest that the consistency of this result across studies helps interpret recent data demonstrating phase coding in Vg neurons during free air whisking (*Wallach et al., 2016*). The work of *Campagner et al. (2016)* shows that during non-contact whisking, a GLM with access to angular acceleration can account for much of the Vg firing. With the assumption that Vg neurons are mechanically sensitive, our analyses suggest that the phase encoding described by *Wallach et al. (2016)* and the angular acceleration tuning described by *Campagner et al. (2016)* both result from inertial forces on the follicle that occur during periods of high angular acceleration (*Boubenec et al., 2012*; *Quist et al., 2014*).

Many Vg neurons are known to respond during both non-contact and contact whisking (*Leiser and Moxon, 2007*; *Szwed et al., 2003*). Here we propose that the encoding of mechanical signals provides a unified explanation for both phase tuning during non-contact whisking and responses during contact. Ultimately, a dynamic model that describes inertial forces during non-contact whisking will be required to verify this hypothesis. It remains unknown how downstream neurons might distinguish Vg spikes that encode phase and hypothetically represent inertial forces from Vg spikes that represent contact forces.

In this light, the results of all four recent studies (*Campagner et al., 2016*; *Quist et al., 2014*; *Wallach et al., 2016*, the present study) provide strong support to the view that Vg neural responses more generally represent the mechanical deformations that occur at the level of the follicle, and that apparent correlations between Vg firing and kinematics are a result of inherent correlations between kinematics and mechanics. This line of evidence suggests that previous results describing the encoding of kinematic variables in the Vg correspond to scenarios characterized by strong correlations between kinematic and mechanical variables. It remains possible that central brain regions take advantage of this inherent correlation to extract behaviorally relevant information about object location or features; there is support from both simulation (*Solomon and Hartmann, 2011*) and behavioral (*Bagdasarian et al., 2013*; *Pammer et al., 2013*) studies indicating that rodents could use a combination of $F_x$ and $M$ to determine the 2D location of a contact point.

## Limitations of the current approach

Our models were unable to reach very high prediction accuracies (median R value = 0.30, max = 0.65); this performance is not as good as might be expected in view of previous evidence that Vg

neuron responses are highly precise and repeatable given identical stimuli (*Bale et al., 2015*; *Jones et al., 2004a*, *2004b*).

We offer four explanations for these seemingly low correlation values.

First, we note that in the present study, R value is only computed during contact, in order to avoid inflation of this statistic due to periods of non-contact when spiking is absent (anesthetized) or sparse (awake). When correlation coefficients were computed to include both periods of contact and non-contact in the awake animal, median R-values increased from 0.27 to 0.47 for kinematic models and from 0.26 to 0.38 for mechanical models. Including periods of non-contact in model evaluation will tend to inflate model performance; any variable that captures transitions between contact and non-contact will easily predict the associated changes in Vg firing rate.

Second, the present work, as well as the majority of reports of Vg neuron firing activity in both awake and anesthetized experiments, is based entirely on a 2D analysis, even though there is ample evidence that the whisker moves in 3D (*Hobbs et al., 2015*, *2016a*; *Huet and Hartmann, 2014*, *2016*; *Huet et al., 2015*; *Knutsen et al., 2008*; *Yang and Hartmann, 2016*) and that Vg neurons are directionally tuned in three dimensions (*Jones et al., 2004a*; *Lichtenstein et al., 1990*; *Minnery and Simons, 2003*).

Third, the quasi-static models used to compute forces and moments at the base of the whisker omit the effects of friction and whisker dynamics, including collisions and vibrations (*Boubenec et al., 2012*; *Jadhav et al., 2009*; *Quist et al., 2014*; *Ritt et al., 2008*; *Wolfe et al., 2008*; *Yan et al., 2013*). To predict spikes at high temporal resolution would require the use of a dynamic model and the ability to track the whisker at spatiotemporal resolutions beyond the capability of the videographic approaches used here.

Lastly, our models are based on linear combinations of stimuli that vary over wide ranges. The only nonlinearity in the model, a static nonlinearity applied to the linear combination as a whole, accounts for the Poisson nature of spiking statistics. This type of simplified Linear-Nonlinear (LNL) model offers strong mathematical advantages; in the case of a GLM, a guarantee that the fitting function that determines the coefficients of the model is convex and has a unique solution easily reachable by gradient methods. However, these models do not allow for linear combinations or non-linearities that could be specific to some regions in the space of inputs. As our experimental methods sample wider regions of input space, it seems reasonable to expect that a single linearized assumption over the full space followed by a single, global nonlinear transformation will prove to be too simplistic. The relatively low quality of prediction achieved here thus might signal the limitations of this type of GLM.

Another limitation of our approach is a time resolution of 15 ms, considerably less than the ms or even sub-ms resolution exhibited by Vg neurons (*Bale et al., 2015*; *Jones et al., 2004a*). Temporal resolution was similarly limited in the study of *Campagner et al. (2016)*, who employed a 100 ms window in contrast to our 15 ms Gaussian kernel. This limit is due in part to experimental constraints in the temporal resolution of the kinematic and mechanical variables chosen as explanatory variables for Vg activity and used as GLM inputs, as addressed in both *Results* and *Materials and methods*. As discussed above, the quasi-static models used to compute forces and moments at the base of the whisker further limit the achievable time resolution.

In addition, both our work and that of *Campagner et al. (2016)* use a similar single-trial modeling approach. Trial averaging would have allowed us to predict spike timing with higher accuracy (*Bale et al., 2013*), but would have required precise duplication of motor command across trials. The variability of whisking behavior in awake animals prevents this duplication. As for the deflection experiments in anesthetized animals, precise duplication could only be achieved by sampling within a narrow region of stimulus space, an approach deliberately avoided here in order to achieve kinematic and mechanical decoupling.

Our work thus offers predictive accuracies as high as can be achieved within these experimental and modeling limitations. The results point towards the conclusion that mechanics more accurately predict primary sensory neuron firing than kinematics, within the hypothesis of linear-non-linear (LNL) encoding, and when the two sets of variables are decoupled. A more stringent test of this hypothesis would require a full 3D characterization of both kinematic and mechanical signals at higher spatiotemporal resolution, a full dynamic model of the whisker for computing forces and moments at its base, and possibly an increased level of modeling sophistication beyond GLMs.

Ultimately, access to a large, decoupled input space is likely to be critical in understanding the coding properties of Vg neurons during natural behavior. Body or head restrained animals tend to generate relatively stereotyped, small angle whisking motions (*Deutsch et al., 2012*) that sample the input space within the coupled regime (*Figure 3*). However, tactile information acquired through whisking during exploratory behavior is varied and complex (*Arkley et al., 2014*; *Carvell and Simons, 1990*; *Grant et al., 2009*; *Hobbs et al., 2016a*, *2016b*; *Mitchinson et al., 2007*; *Saraf-Sinik et al., 2015*; *Schroeder and Ritt, 2016*; *Sellien et al., 2005*; *Towal and Hartmann, 2008*; *Voigts et al., 2015*). Neurons of the Vg must be able to encode the signals associated with the full range of potential stimuli, including large angle deflections and very distal contacts. By adopting a mechanical characterization of tactile information, we can quantify the large input space available during tactile sensation in a manner that incorporates the true shape and deformability of the whisker.

## Materials and methods

All procedures involving animals were approved in advance by the Northwestern University Animal Care and Use Committee. A total of fourteen female Long Evans rats (age 2–6 months) were used.

### Surgical procedures

Animals were anesthetized with a ketamine-xylazine hydrochloride combination delivered intraperitoneally (60 mg/kg ketamine, 3.0 mg/kg xylazine, and 0.6 mg/kg acepromazine maleate). Four or five stainless steel screws were placed in the skull over neocortical areas and covered in dental acrylic. For anesthetized recordings this structure was affixed to the surgical bed; for chronic (awake) recordings, it formed the base of the electrode implant.

A small (~1 mm diameter) craniotomy was then performed in order to allow access to the trigeminal ganglion (Vg), at location ~2 mm caudal relative to bregma and ~2 mm lateral to the midline. A single tungsten electrode (FHC, Bowdoin, ME; typical impedance 2–5 MΩ) was lowered to a depth of ~10 mm until multi-unit responses to whisker deflections could be heard. The electrode was then lowered more slowly until isolated single neuron responses to tactile stimulation of a single vibrissa were obtained.

For chronic recordings, the electrode was then fixed in place using dental acrylic. In some animals, electrodes were bilaterally implanted in the Vg. Recordings from awake, chronically implanted animals were started no sooner than four days after surgery and continued for up to three weeks. All chronic implantation surgeries were performed in a sterile field.

### Anesthetized recordings

Five animals were used to test the responses of Vg neurons to passive, manual deflection. After performing the craniotomy described above, single tungsten electrodes (FHC ~1 MΩ) were lowered to a depth of ~10 mm until a neuron that responded to the deflection of a single whisker was isolated. We recorded video from a top-down view at 300 fps with an exposure time of 1 ms (Teledyne Dalsa Genie HM640; Waterloo, Canada).

Neural signals were amplified on an A-M Systems (Sequim, WA) four channel amplifier (1000x gain) with analog bandpass filtering between 10 Hz and 10 kHz before digital sampling at 40 kHz using Datawave SciWorks (Loveland, CO). After acquisition, traces were digitally bandpass filtered between 300 Hz and 6000 Hz before spike sorting. Spikes were identified and sorted offline, and spike times were rounded to the nearest ms for comparison with video data. Examples of raw data are shown in *Figure 1A–C*.

In order to robustly track the whisker in the high-speed video, the surrounding fur was removed with depilatory cream (Nair; Church and Dwight, Ewing, NJ) and surrounding whiskers were either trimmed or held back against the fur. Care was taken not to deform the whisker or the mystacial pad during recordings.

Whiskers were deflected manually by pressing a 0.3 mm graphite rod against the whisker (*Video 1B*). Between 20 and 40 deflections were applied at variable radial distances (up to 90% of the whisker length), at two velocities and two directions (rostral to caudal, and caudal to rostral) for a total of 80–160 deflections per whisker. Analyses were restricted to distal contacts (>40% of the whisker length), where the follicle does not move appreciably during contact. Whiskers were also

held in a deflected position for periods of about 3 s to test adaptation characteristics. All deflections were on the order of several mm.

## Awake recordings

Seven animals were gentled for 8–10 days prior to surgery. During gentling, rats were acclimated to restraint in a V-shaped fabric bag that prevented body movement but permitted head and neck movements.

Starting four days after surgery, on each day of testing we gently restrained the rat and again confirmed that each neuron responded to tactile stimulation of one and only one whisker. All other whiskers on that side of the rat's face were trimmed to the level of the fur. Rats were then placed in the fabric bag, and high-speed video (Photron FastCam, San Diego, CA; either 1024PCI or 512PCI) was used to record the top-down view of the rat's head as it whisked against a rigid vertical peg (3 mm diameter). Video was taken at 1000 fps, with a shutter speed of 1/3000 s to reduce motion blur.

Signals from Vg neurons were recorded with a Triangle Biosystems (Durham, NC) 8-channel pre-amplifier (2x gain) and a custom-built amplifier (500x gain). Signals were analog band-pass filtered between 0.33 Hz and 10 kHz before sampling at 40 kHz using Datawave SciWorks. Traces were then digitally bandpass filtered between 300 Hz and 8000 Hz before spike sorting. Spikes were identified and sorted offline, and spike times were rounded to the nearest ms for comparison with video data. Examples of raw data are shown in *Figure 1D–E*.

## Calculation of kinematic and mechanical variables

For both anesthetized and awake experiments, whisker shape was extracted from each video frame using the software "Whisk" (*Clack et al., 2012*). The kinematic and mechanical variables of contact were computed from the whisker shape; see *Figure 2* of Results.

The kinematic variables of contact are: radial distance ($r$), angle of contact ($\theta_{push}$ or $\theta_{deflection}$), and angular velocity ($V$). The variable $r$ is the linear distance between the basepoint and the contact point. The variable $\theta_{deflection}$ is valid for manual deflection; as illustrated in *Figure 2A*, it is the angle between two line segments: one that connects the initial point of contact to the whisker basepoint and one that connects the current contact point to the whisker basepoint (*Gibson and Welker, 1983a*, *1983b*; *Lichtenstein et al., 1990*; *Lottem and Azouz, 2009*, *2011*; *Shoykhet et al., 2000*, *2003*). The variable $\theta_{push}$ is valid for active whisking; as illustrated in *Figure 2D*, it represents the angle swept out by the tangent to the whisker at its base from the time of contact onset to the current time (*Bagdasarian et al., 2013*; *Quist and Hartmann, 2012*; *Solomon and Hartmann, 2011*). The velocity ($V$) is the temporal derivative of either $\theta_{deflection}$ or $\theta_{push}$.

The mechanical variables of contact are the axial force ($F_x$), the force parallel to the whisker axis near its base, positive pointing out of the follicle; the transverse force ($F_y$), the force perpendicular to the whisker axis, directed in the rostral direction; and the bending moment ($M$), the moment about the vertical z-axis that passes through the whisker base. Mechanical variables were computed using a quasi-static model of whisker bending (*Birdwell et al., 2007*; *Quist and Hartmann, 2012*; *Solomon and Hartmann, 2008*, *2010*).

All mechanical and kinematic data were median filtered to eliminate point outliers. Variables computed from video acquired at 300 fps were linearly interpolated to 1000 Hz for comparison with spike times on the 1 ms scale. Velocity was calculated using a central difference approximation of the angular component of contact and low pass filtered at 85 Hz.

The spike train was smoothed with a Gaussian kernel with standard deviation $\sigma$ to find the rate $r(t)$:

$$r(t) = \frac{1}{\sqrt{2\pi\sigma^2}} \sum_{j=1}^{N} e^{\frac{-(t-t_j)^2}{2\sigma^2}}, \tag{1}$$

where $N$ is the total number of spikes, $\sigma$ is the standard deviation of the kernel, and $t_j$ is the time of spike $j$. The standard deviation $\sigma$ of the Gaussian kernel was varied between 1 ms and 500 ms to observe the effect of temporal smoothing on the quality of predictions. An optimal kernel width of $\sigma = 15$ ms was chosen for all subsequent analyses. This was the smallest value of $\sigma$, below which we observed a sharp decrease in the quality of predictions.

## Generalized linear models

Each GLM is of the form:

$$p(t) = f\left(\sum_{j=1}^{K} \sum_{t'=0}^{\tau} \alpha_j(t')\, x_j(t-t')\right). \tag{2}$$

Here $p(t)$ is the probability that the neuron emits a spike in the 1 ms time interval centered at time $t$, $f$ is a logistic nonlinearity, and $j$ sums over all the predictor variables. Each one of these variables $x_j$, $1 \leq j \leq K$, contributes to the argument of the logistic nonlinearity through its current value and its values in the preceding $\tau$ time bins, weighted by the filter parameters $\alpha_j(t')$, $0 \leq t' \leq \tau$. Full models used ($r$, $\theta$, $V$, $F_x$, $F_y$, $M$) as predictor variables ($K=6$), while subset models had access to either kinematic variables ($r$, $\theta$, $V$) or mechanical variables ($F_x$, $F_y$, $M$), so that $K=3$.

Since the neural response is quantified as a spike either present or absent in each 1 ms time bin, the statistics process being modeled is Bernoulli and the nonlinearity is sigmoidal (**McCullagh and Nelder, 1989**):

$$f(u) = \frac{1}{1+e^{-u}}. \tag{3}$$

The GLM finds the filters $\{\alpha_j(t')\}$, $0 \leq t' \leq \tau$, $1 \leq j \leq K$ that maximize the likelihood of the observed spiking activity. To enforce continuity of the filters as a function of time and reduce the number of coefficients needed to specify the model, it is convenient to introduce a basis of 'raised cosine bumps' $b_l(t)$, $1 \leq l \leq L$ (**Pillow et al., 2008**). Here we used the $L=5$ basis shown in **Figure 4A**. The functions peak at 0 ms ($l=1$), 1 ms ($l=2$), 3 ms ($l=3$), 8 ms ($l=4$), and 17 ms ($l=5$); the basis covers 75 ms into the past.

The expansion of each filter in terms of this basis, namely

$$\alpha_j(t') = \sum_{l=1}^{L} b_l(t')\, \beta_{lj}, \tag{4}$$

results in an interesting reformulation of the GLM:

$$p(t) = f\left(\sum_{j=1}^{K} \sum_{l=1}^{L} \beta_{lj}\, \tilde{x}_{lj}(t)\right), \tag{5}$$

where the input variables to the model $\tilde{x}_{lj}(t)$ are now 'convolved predictors,' the filtered versions of the original input variables, namely:

$$\tilde{x}_{lj}(t) = \sum_{t'=0}^{\tau} b_l(t')\, x_j(t-t'). \tag{6}$$

In this formulation, the problem of fitting the parameters of the GLM is reduced from that of finding the filters $\alpha_j(t')$, $1 \leq j \leq K$, $0 \leq t' \leq \tau$, to that of fitting a smaller number of parameters: the coefficients $\beta_{lj}$, $1 \leq l \leq L$, $1 \leq j \leq K$.

To evaluate each GLM we implemented ten-fold cross-validation, using 90% of each neuron's dataset to fit the coefficients $\beta_{lj}$. The fitted GLM was used to predict the spike rate on the remaining 10% of the data. This procedure was repeated ten times, so that the entire neural response was eventually predicted from a model whose coefficients were fit on independent data. This method prevents overfitting and allows the model to be evaluated based on how well it generalizes to new data.

The quality of each GLM was quantified through the correlation coefficient between the predicted rate $p(t)$ of **Equation 5** and the rate $r(t)$ obtained from **Equation 1**.

Data for non-contact periods were omitted in calculations of correlation coefficients. Given that Vg neurons do not fire during non-contact, a precise prediction of no activity during these periods would have unduly inflated model performance. Predictions were tracked only during contact periods.

The percent difference between the subset models and the full model was calculated as $100 * \frac{\left(R_{full} - R_{subset}\right)}{R_{full}}$ where $R$ is the Pearson Correlation Coefficient between the observed spike rate $r(t)$ and the predicted spike rate $p(t)$ obtained with either the full model or one of the two subset models.

## Additional information

### Funding

| Funder | Grant reference number | Author |
|---|---|---|
| National Science Foundation | IOS-0818414 | Mitra JZ Hartmann |
| National Science Foundation | IOS-0846088 | Mitra JZ Hartmann |
| National Science Foundation | EFRI-0938007 | Mitra JZ Hartmann |
| National Institute of Neurological Disorders and Stroke | R01-NS093585 | Mitra JZ Hartmann Sara A Solla |
| National Institutes of Health | T32-HD0578 | Nicholas E Bush |
| National Institutes of Health | F31-NS092335 | Nicholas E Bush |
| National Science Foundation | DGE-0903637 | Christopher L Schroeder |
| Air Force Office of Scientific Research | 32 CFR 168a | Lucie A Huet |

The funders had no role in study design, data collection and interpretation, or the decision to submit the work for publication.

### Author contributions

NEB, CLS, Conception and design, Acquisition of data, Analysis and interpretation of data, Drafting or revising the article; JAH, AETY, LAH, Analysis and interpretation of data, Drafting or revising the article; SAS, MJZH, Conception and design, Analysis and interpretation of data, Drafting or revising the article

### Author ORCIDs

Mitra JZ Hartmann, http://orcid.org/0000-0003-0783-1483

### Ethics

Animal experimentation: All procedures involving animals were approved in advance by the Northwestern University Animal Care and Use Committee protocols #2012-1776 and #2015-1575.

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
