## [Decision Letter]

Thank you for submitting your work entitled "Decoupling kinematics and mechanics reveals coding properties of trigeminal ganglion neurons in the rat vibrissal system" for consideration by *eLife*. Your article has been reviewed by two peer reviewers, and the evaluation has been overseen by David Kleinfeld as the Reviewing Editor and David Van Essen as the Senior Editor.

The reviewers have discussed the reviews with one another and the Reviewing Editor has drafted this decision to help you prepare a revised submission.

David Kleinfeld has also offered a comment on the comparison between your work and that of Campagner et al.

We ask that you perform these changes before a decision is made to accept or reject the manuscript:

1) The Reviewing Editor requests examples of stimulus triggered averages of the spike response, on a high resolution scale, should be added to Figure 1.

2) Add a thoughtful and convincing discussion, possibly with reanalysis of data as needed, comparing the current work with that by Campagner et al.

3) Further to the above point, convincingly address the critical issue raised by Reviewer #2 regarding differences with the study by Campagner et al.: "The result here is opposite: mechanical variables predict the firing of Vg neurons better than kinematic variables for passive deflection but not for active touch. I do not understand this result, but it may be an outcome of the limitation of the GLM approach (see subsection ‘3.3. Limitations of the current approach’, and the low values of R in Figure 6)."

4) Address all points of the two reviewers.

Reviewer #1:

This is a technically sound manuscript. The central finding, that primary sensory neurons in the whisker system encode mainly mechanical variables (forces and moments) rather than kinematic variables (speed, etc.) is important. The approach, based on a Generalized Linear Model analysis of primary afferent responses, is quantitative and rigorous.

Unfortunately, a very similar set of results has just been published in *eLife* by the Petersen group (Campagner et al., 2016). Using recordings from primary neurons of awake mice (rather than rats), these authors related neuronal responses to whisker kinematics and forces. To do so they also adopted a Generalized Linear Model approach, as in Campagner et al. [NB – GLMs were used by the Petersen group in earlier work as well (Bale et al., 2013).] The conclusions of Campagner et al. are very similar to the present manuscript. Primary sensory neurons in the whisker system (of awake animals) encode primarily rotational forces, rather than whisker angle or speed. The Campagner study also includes a comparison of active vs. passive stimulation. In the present form of the Bush et al. manuscript, there are no distinctive additional or broader findings as compared to Campagner et al.

There is a difference between the papers in the analysis, whereby in Bush et al. mechanical variables predict neuronal firing better than kinetic variables during manual stimulation but not active touch, whereas in Campagner et al. mechanical variables are better predictors during active touch. The two types of variables decouple better during manual stimulation in Bush et al., but during active touch in Campagner et al. This is likely to be an effect of differences in experimental design and stimulation parameter range. GLM fit quality falls within a similar range for both papers.

Because the present version of the submitted manuscript reaches very similar conclusions (albeit in rats, not mice) and the main methodological contribution is again similar (use of GLMs), unfortunately I must recommend against the manuscript's acceptance in *eLife* in its present form.

Reviewer #2:

Bush et al. quantify how well responses of Vg neurons in the vibrissa system are explained by either mechanical signals at the whisker base (*F_x_, F_y_* and *M*) or by kinematic variables (the push or deflection angles *θ*, the point of contact *r* and the angular velocity *V*; acceleration is not considered in contrast to previous studies). For that purpose, they develop a GLM statistical model. They find that Vg neurons more directly encode mechanical signal when the whisker is deflected, but there is no evident preference for kinematic or mechanical models during active touch.

The mechanoreceptors in the follicle are sensitive to their immediate surrounding. In the past, the last author (and other) has studied bending properties of whiskers that are firmly held under the quasi-static approximation. They have shown that the variables *F_x_, F_y_* and *M* can be calculated if *θ* and *r* are known, and vice versa. For that case, the information about the firing rates of Vg neurons from the mechanical variables should be equal to the information extracted from the kinematic variables. The whisker, however, emerges from the mystacial pad and the whisker base can move under the effect of contact forces. Therefore, kinematic variables do not determine the forces uniquely if the properties of the pad vary with time. Forces and moments that directly affect the mechanoreceptors are expected to predict the firing properties of Vg neurons better than kinematic variables.

I expect that for anesthetized animals, the properties of the mystacial pad will not vary with time in a considerable manner. In awake animals, the motor system of animal may change properties of the pad. Therefore, the kinematic and mechanical variable would yield more similar results for anesthetized animals than for awake animals. This is the result of an *eLife* article that was published on Feb. 15 by Campagner et al. (and therefore the authors could not know about it when they submitted the present manuscript), although there are differences between the two studies (for example, Campagner et al. consider only the whisker angle, and not *r* and *V*). The result here is opposite: mechanical variables predict the firing of Vg neurons better than kinematic variables for passive deflection but not for active touch. I do not understand this result, but it may be an outcome of the limitation of the GLM approach (see subsection “3.3. Limitations of the current approach”, and the low values of R in Figure 6).

One may consider publishing this manuscript to contrast it with the manuscript of Campagner et al. In an era when scientists care about "reproducibility", publishing the two papers together may hint about the problematic of statistical techniques like the GLM approach.

Comment:

Figure 3, left: *F_x_* values are widely distributed for the same *θ_push_*. In particular, they are sometimes positive and sometimes negative for the same *θ_push_*. How can it be?

Editor’s comment (David Kleinfeld):

The recent data by Petersen's group (Campagner et al., *eLife*, 2016) and the new data from Hartmann's group concern coding at the level of primary sensory neurons and how features extracted at this level remain valid up through the level of cortex. Both laboratories show that torque is an essential variable in the coding of contact by primary sensory neurons. Hartmann and colleagues further show that the impulse and the radial location of touch are coded, an issue not considered by Petersen's group, albeit these play a lesser role in determining the spike response. Petersen's group shows that phase in the whisk cycle is encoded in the absence of touch, an issue not explored by Hartmann. Thus the two papers agree on the big issue, perhaps not in every detail, and otherwise offer complementary information on the potential coding of vibrissa contact.

---

## [Author Response]

*We ask that you perform these changes before a decision is made to accept or reject the manuscript:*

*1) The Reviewing Editor requests examples of stimulus triggered averages of the spike response, on a high resolution scale, should be added to Figure 1.*

We have provided the requested examples as a Figure 4—figure supplement 1. In reviewing the overall structure of the manuscript, we found this material to be more appropriate as a supplement to Figure 4 rather than to Figure 1, and hope the editor will agree with this placement.

*2) Add a thoughtful and convincing discussion, possibly with reanalysis of data as needed, comparing the current work with that by Campagner et al.*

We have thoroughly revised the Discussion section of our manuscript to include a comparison of our work to that of Campagner et al. The changes were sufficiently extensive that we have redlined the heading “Discussion” instead of all the individual changes in that section of the manuscript.

*3) Further to the above point, convincingly address the critical issue raised by Reviewer #2 regarding differences with the study by Campagner et al.: "The result here is opposite: mechanical variables predict the firing of Vg neurons better than kinematic variables for passive deflection but not for active touch. I do not understand this result, but it may be an outcome of the limitation of the GLM approach (see subsection ‘3.3. Limitations of the current approach’, and the low values of R in Figure 6)."*

We have addressed this question in the revised Discussion section of our paper; see also our response to reviewer #2.

4) Address all points of the two reviewers.

Please see below.

*Reviewer #1:*

This is a technically sound manuscript. The central finding, that primary sensory neurons in the whisker system encode mainly mechanical variables (forces and moments) rather than kinematic variables (speed, etc.) is important. The approach, based on a Generalized Linear Model analysis of primary afferent responses, is quantitative and rigorous.

We thank the reviewer for this positive evaluation of our manuscript.

Unfortunately, a very similar set of results has just been published in eLife by the Petersen group (Campagner et al., 2016). Using recordings from primary neurons of awake mice (rather than rats), these authors related neuronal responses to whisker kinematics and forces. To do so they also adopted a Generalized Linear Model approach, as in Campagner et al. [NB – GLMs were used by the Petersen group in earlier work as well (Bale et al., 2013).] The conclusions of Campagner et al. are very similar to the present manuscript. Primary sensory neurons in the whisker system (of awake animals) encode primarily rotational forces, rather than whisker angle or speed. The Campagner study also includes a comparison of active vs. passive stimulation. In the present form of the Bush et al. manuscript, there are no distinctive additional or broader findings as compared to Campagner et al.

A number of important differences distinguish the present work from that of Campagner et al.

The most significant difference between the two studies is the choice of variables to represent kinematic and mechanical information. Given that Vg neurons are far more likely to fire during contact than during non-contact, any variable chosen to explain Vg firing – whether geometric or mechanical – should carry contact information.

Campagner et al. employ a mechanical variable (curvature change) that carries significant information about contact vs. non-contact, but employ a geometric variable (*θ_head_*) that carries no information about contact vs. non-contact. It is therefore evident that curvature change will be a better predictor of Vg firing than *θ_head_*. The Campagner et al. finding that mechanical variables are better predictors of Vg firing than kinematic variables is attributable to an inappropriate choice of kinematic variable.

The present work also favors mechanical over geometric variables as predictors of Vg firing, but we have employed geometric variables that contain information about contact vs. non-contact. The angular variables *θ_push_*(during active whisking) and *θ*_deflection_ (during passive deflection) are nonzero only during contact, as are their mechanical counterparts. Thus, in the present work, the comparison of the relative predictive power of kinematic and mechanical variables is unbiased.

A second important difference between the results presented by Campagner et al. and our results arises from the approach used to compute the Pearson Correlation Coefficient (R). During periods of non-contact, firing rates of most Vg neurons are very low. Both the models proposed by us and those proposed by Campagner et al. are very good at predicting the quiet periods in Vg activity during non-contact. We decided to quantify the goodness of our models by focusing on their ability to predict Vg firing during contact; thus we only used firing data during contact to compute R values. By using firing data during both contact and non-contact to compute R values, Campagner et al. magnify the prediction accuracy of their models.

A third difference between the two studies is that Campagner et al. construct models based on a single explanatory variable for Vg firing; they choose curvature change for mechanics and *θ_head_*for kinematics. These single-input models neglect contributions from other components of kinematics (radial distance to object, velocity) and mechanics (*F_x_*) that have been demonstrated to be important not only for explaining the response of Vg neurons (Stüttgen et al. 2008; Lottem et al., 2015) but also for behavior (Pammer et al., 2013). Our work takes into account these additional inputs.

Fourth, Campagner et al. claim to have “measured forces” or “calculated forces,” when in actuality they measure a geometric quantity – curvature change at the base – and use it as a proxy for bending moment. This proxy relies on the accurate determination of base curvature, which is highly susceptible to tracking errors and can easily become inaccurate. In our work, we employ a validated mechanical model that uses a quasi-static approximation to quantify the forces and moments applied to the whisker base (Solomon and Hartmann, 2006; Quist et al., 2012; Huet et al., 2015; Huet and Hartmann, 2016).

Finally, the results reported here indicate that mechanics predict Vg firing better during passive stimulation than during active contact, while Campagner et al. show the exact opposite. As indicated by Reviewer 2, the two studies thus appear to directly contradict. We offer an explanation for this apparent contradiction: the discrepancy is not related to active vs. passive touch, but rather to whether the mechanics and the geometry are decoupled in a given scenario.

There is a difference between the papers in the analysis, whereby in Bush et al. mechanical variables predict neuronal firing better than kinetic variables during manual stimulation but not active touch, whereas in Campagner et al. mechanical variables are better predictors during active touch. The two types of variables decouple better during manual stimulation in Bush et al., but during active touch in Campagner et al. This is likely to be an effect of differences in experimental design and stimulation parameter range. GLM fit quality falls within a similar range for both papers.

The apparent discrepancy between the two works has been addressed in detail in the revised Discussion section.

We attribute the differences primarily to the inappropriate choice of *θ_head_* as the kinematic predictor in Campagner et al. As discussed above, *θ_head_* carries no information about contact; it varies significantly during a whisk regardless of whether contact occurs. In contrast, curvature changes will only occur during contact. If contact takes place at different positions with respect to the head, it is evident that *θ_head_* and curvature changes will decouple. However, in the experimental design used by Campagner et al., passive displacement always occurs at the same angular position; this prevents the decoupling between *θ_head_* and curvature changes. In contrast, an experimental design in which passive stimulation was applied at a variety of angular positions would allow curvature changes and *θ_head_* to decouple. Campagner et al.did not explore this scenario, while our experimental approach to passive deflection was specifically designed to uncouple kinematic from mechanical variables.

Because the present version of the submitted manuscript reaches very similar conclusions (albeit in rats, not mice) and the main methodological contribution is again similar (use of GLMs), unfortunately I must recommend against the manuscript's acceptance in eLife in its present form.

In our arguments above and in the revised Discussion session we argue that there is a flaw in the choice of kinematic variable by Campagner et al. – the choice of *θ_head_* makes it a foregone conclusion that the mechanical variable will have a larger ability to predict Vg firing during active whisking. In addition, their failure to decouple kinematic from mechanical variables during passive deflection leads them to misattribute the difference to awake vs. passive conditions.

*Reviewer #2:*

Bush et al. quantify how well responses of Vg neurons in the vibrissa system are explained by either mechanical signals at the whisker base (F_x_, F_y_ and M) or by kinematic variables (the push or deflection angles θ, the point of contact r and the angular velocity V; acceleration is not considered in contrast to previous studies). For that purpose, they develop a GLM statistical model. They find that Vg neurons more directly encode mechanical signal when the whisker is deflected, but there is no evident preference for kinematic or mechanical models during active touch.

We thank the reviewer for this accurate summary of our work.

*The mechanoreceptors in the follicle are sensitive to their immediate surrounding. In the past, the last author (and other) have studied bending properties of whiskers that are firmly held under the quasi-static approximation. They have shown that the variables F_x_, F_y_ and M can be calculated if θ and r are known, and vice versa. For that case, the information about the firing rates of Vg neurons from the mechanical variables should be equal to the information extracted from the kinematic variables.*

We agree with the reviewer (only *M* and *F_x_*are needed to compute *θ* and *r*), however, we note that the relation between mechanical and kinematic variables is significantly nonlinear. The present work adopts a simple model of neural encoding: that Vg neurons respond to a linear combination of relevant features of the stimulus, followed by a static nonlinearity that simply accounts for the Poisson statistics of the spike generation process. This is the conceptual framework that underlies the choice of GLM models. In asking which set of variables, kinematic vs. mechanical, are better predictors of Vg activity when used as inputs to a GLM model, we are asking which set of variables is more informative within the hypothesis of linear-non-linear (LNL) encoding.

*The whisker, however, emerges from the mystacial pad and the whisker base can move under the effect of contact forces. Therefore, kinematic variables do not determine the forces uniquely if the properties of the pad vary with time. Forces and moments that directly affect the mechanoreceptors are expected to predict the firing properties of Vg neurons better than kinematic variables.*

I expect that for anesthetized animals, the properties of the mystacial pad will not vary with time in a considerable manner. In awake animals, the motor system of animal may change properties of the pad. Therefore, the kinematic and mechanical variable would yield more similar results for anesthetized animals than for awake animals. This is the result of an eLife article that was published on Feb. 15 by Campagner et al. (and therefore the authors could not know about it when they submitted the present manuscript), although there are differences between the two studies (for example, Campagner et al. consider only the whisker angle, and not r and V). The result here is opposite: mechanical variables predict the firing of Vg neurons better than kinematic variables for passive deflection but not for active touch. I do not understand this result, but it may be an outcome of the limitation of the GLM approach (see subsection “3.3. Limitations of the current approach”, and the low values of R in Figure 6).

The differences between the current work and the Campagner et al. work have been addressed in detail in the response to Reviewer 1 (see above), as well as in the revised Discussion section.

Contrary to how it may appear, whether the whisker deflections are active/passive is not what determines the relative ability of mechanical vs. geometric variables to predict Vg firing. Instead, it is whether we consider Vg responses in a regime in which mechanical and geometric variables are decoupled.

As the reviewer suggests, properties of the mystacial pad can, under some conditions, contribute to this decoupling, but these properties are not the sole source of decorrelation between kinematics and mechanics. The intrinsic curvature, deformability, and varying stiffness of the whisker all contribute to a complex relationship between kinematics and mechanics.

As detailed in the Discussion section, standard approaches typically use small angle passive stimulation very close to the whisker base. In this region the whisker is stiff and behaves essentially like a rigid body. The present work relaxes the rigid body assumptions and addresses how curvature, deformability, and stiffness affect kinematics and mechanics through a novel stimulation protocol. This approach allows us to correlate neural responses to a much wider and more naturalistic range of input stimuli, one that has not been tested before.

To summarize an argument now presented in the revised Discussion section: Campagner et al. use the kinematic variable *θ_head_*, which is measured with respect to the midline of the animal’s head. They compare this to curvature change, which is measured independently of the whisker’s position relative to the head. These variables will be decoupled in the awake experiments because contact with an object can occur at different positions relative to the head. The same protraction against a peg located for instance, at 70° and at 110°, will exhibit very similar curvature changes but at very different values for *θ_head_*. In contrast, in the Campagner et al. anesthetized experiments, *θ_head_* and curvature are always strongly correlated because the whisker is trimmed (to 5 mm), angles of deflection are relatively small (10°), and the deflection is always applied at the same *θ_head_*. Had they used large amplitude deflections and/or deflections further from the whisker base, *θ_head_* would presumably have decoupled from curvature changes.

In our experiments, we define both kinematics and mechanics independently of the whisker’s position relative to the head. In our awake experiments, it is less common and less reliable to observe decoupling of kinematics from mechanics. In contrast, the design of our deflection protocol makes it much more likely that kinematics and mechanics will be decoupled in our anesthetized experiments.

It is a combination of an appropriate choice of variables and a different design of our experiments in the anesthetized animal that accounts for the apparent difference between the two works.

One may consider publishing this manuscript to contrast it with the manuscript of Campagner et al. In an era when scientists care about "reproducibility", publishing the two papers together may hint about the problematic of statistical techniques like the GLM approach.

We appreciate the reviewer’s recommendation, and refer to our revised Discussion section, which now includes an extensive comparison of the work of Campagner et al. with our work and specifically discusses the apparent contradictions between the two studies.

*Comment:*

*Figure 3, left: F_x_ values are widely distributed for the same θ_push_. In particular, they are sometimes positive and sometimes negative for the same θ_push_. How can it be?*

Indeed, *F_x_* is sometimes positive and sometimes negative for the same *θ_push_*. This effect occurs because of basepoint translations, head movements, and changes in whisker orientation (concave forward or concave backward) during *θ_push_*. More specifically, when a concave-forward whisker protracts against a peg, the protraction will tend to pull the whisker out of the follicle; *F_x_* is thus positive. When a concave-backwards whisker protracts against the peg, the protraction will tend to push the whisker into the follicle, so *F_x_* is negative.

We thank the reviewer for this critical comment about Figure 3. Our reexamination of this data brought to light a minor error in our calculation of geometric variables used across analyses. This error was identified and corrected, and all data were re-analyzed. This resulted in changes in Figure 3 to 6; notably, these corrections did not alter any of the findings reported in this manuscript or their significance.

Note that we now show a different neuron in Figure 3. The reason for this choice is that the purpose of this figure is to demonstrate that kinematics and mechanics are not necessarily tightly coupled in the awake animal. After correcting our data, we found that a different neuron better exemplified this effect.